# Market integration reduces kin density in women's ego-networks in rural Poland

Heidi Colleran 1*

It is widely assumed that as populations become more market integrated the 'inner circles' of people's social networks become less densely connected and family-oriented. This 'loosening' of kin networks may fundamentally alter the social dynamics of reproduction, facilitating demographic transitions to low fertility. Few data exist to test this hypothesis. Previous research in urbanized populations has not explicitly measured kin density in ego-networks, nor assessed how market integration influences network structure at different levels of aggregation. Here I analyze the ego-networks of ~2000 women in 22 rural Polish communities transitioning from subsistence farming to market-dependence. I compare how ego-network size, density and kin density co-vary with household and community-level market integration. Market integration is associated with less kin-dense networks, but not necessarily less dense ones, and is unrelated to network size. Declining kin density during economic transitions may be a critical mechanism for the broader cultural transmission of low fertility values.

---

[1] BirthRites Independent Research Group, Department of Human Behavior, Ecology and Culture, Max Planck Institute for Evolutionary Anthropology, Deutscher Platz 6, 04103 Leipzig, Germany. *email: heidi_colleran@eva.mpg.de

The question of how 'community' is transformed as populations go through major economic and social changes is foundational to many social sciences[1,2]. For most of human history, kinship has been the fundamental structuring principle for economic, socio-political and ritual relations, but it is widely assumed that 'traditional' social interactions break down as populations become more market integrated[3–6]. Specifically, personal networks are assumed to become less kin-oriented and densely connected. This 'loosening' of kin relations, as part of a suite of profound social and structural changes during the course of market integration, is thought to be a critical mechanism driving the global demographic transition to low fertility.

Sociologists have long examined whether kinship, friendship and neighbourhood ties are 'lost', 'saved' or 'liberated' in the urbanisation process[7], and whether declining kin interactions are a cause or a consequence of industrial intensification[8–10]. This research is dominated by evidence from urban populations in Western Europe and North America[6,11–15]. A parallel literature in anthropology has focused on how social networks in 'small-scale' and mixed subsistence economies compare to those in market-dependent ones[16–19]. This work has focused more on how networks maintain cultural complexity[20–22], reproduce social and political equilibria[23], buffer food and other shortages[18,24], signal reputation[25] and social capital[26] but less on what happens to those networks in times of transition.

All economic transitions are associated with profound demographic and socio-cultural changes that affect both the availability of kin as network partners and the costs and benefits of different social structures. It is important to situate network studies within the demographic context of the communities undergoing change, because these features co-evolve. For example, high mobility among some hunter-gatherer groups can generate relatively low relatedness and apparently less kin-oriented networks[17,27] than among sedentary farmers. Transitions to agriculture can generate increasingly kin-saturated social worlds because of extended family co-residence, greater reliance on food storage, higher fertility[27] and mortality, leading to larger, denser, more interrelated populations[17,28–31]. Transitions away from subsistence farming to a market economy are similarly associated with major demographic changes: relatively fewer complex family living arrangements, higher mobility—both geographic and social —associated with wage-labour and education, lower mortality due to improvements in sanitation and infrastructure, and declining fertility due to decreased reliance on primary agricultural productivity and a parallel increase in human capital investment[3,32–35].

Declining density of social networks in the course of these transitions has significant implications for how resources and social support flow between individuals, for the content of interpersonal exchanges and how this influences cultural change, and ultimately for the demography of a population. Dense networks can generate social interdependence and rapid consensus-formation but also social control and resistance to change[36,37] via similar mechanisms, such as conformity and social influence. Tight-knit relationships can help people monitor and maintain social norms, but intimate groups also have their own modes of discourse, reference, and interpretation of the world and so a dense network can be insular and socially cut-off. Low-density networks instead evoke the idea of transient, untethered, transactional or impersonal social contacts. Sparser networks with diverse, 'weak'[38] and cross-cutting connections can potentially spread novel information easily and quickly within a community. This can facilitate rapid adaptation to social and environmental challenges, help people reject existing social hierarchies, and accumulate cultural innovations across communities through partial connectivity[21].

Whether these inner circles are composed of kin or non-kin is important because kin can be powerful brokers of behaviour change, blocking or backing new ideas, often while having vested interests in the domain of reproduction. Evolutionary anthropologists tend to assume that humans evolved as 'cooperative breeders'[39], relying on kin for support of reproduction[40]. Assuming that kin have broadly pro-natal or fitness-consistent influences on reproduction[41], increasing non-kin interactions could disrupt these evolved patterns of coordination[42]. Scaling up, by increasing population-level rates of horizontal (peer-to-peer) relative to vertical (parent-to-child) transmission[43], or through weak ties[38], non-kin interactions may allow the spread of new values associated with low fertility in a population[42]. For this reason, women's personal networks have come into focus for research on fertility decline[36,44]. Studies of contraceptive uptake[45–50], drawing on the diffusion of innovations literature[51], have examined how network density either critically slows-down or accelerates the uptake of novel contraceptives and fertility decline[48,49]. Kin have been shown to have important influences on contraceptive use[46,47,52–54], and my colleagues and I[46] have previously argued that personal discussion networks may be an important locus of innovation and experimentation in which community-level norms can be safely violated. Prestige dynamics, conformity and other psychological biases can then help spread innovative behaviour through wider networks and populations.

In urbanised market economies, core network partners change over time and as people go through major life transitions such as marriage and divorce[55–58], but kin are still prominent in social networks. Women's networks typically contain more kin than do men's[6,37,59–61] and married women's networks are the most kin saturated[59]. This could be driven by unequal access to economic and social opportunities for women compared to men[59], gendered domestic and parenting roles that concentrate kin in married women's networks, and time constraints on women's network-formation during reproductive-aged years. In urban USA, China and southern France[14,60,61], personal discussion networks have recently shrunk, become more diverse and less dense. In the US and French cases, kin connections were better maintained over time, while in China more kin connections were lost[61]. Among young adults transitioning from school to university in the UK, kin connections were more stable than non-kin connections, i.e. less sensitive to decay, replacement and abandonment over time[62]. In Canada between the 1970s and 2000s, face-to-face interaction with kin and close friends changed little, and telephone contact increased, with kin being contacted more than non-kin[63]. Because economic modernisation provides the means to stay in contact (i.e. communication systems) and interact (e.g. transport facilities)[10,63,64], geographic mobility does not have to lead to fragmented kin networks[64] and indeed, kin relations appear to rely less on geographic proximity than do non-kin relations[65]. The evidence therefore suggests that in market economies where non-kin are frequently interacted with, kin remain highly represented in personal networks[5,8,14,37,60,66,67], especially among highly educated people[60]. Even where the relationship is not considered intimate, kin are seen as reliable and likely to respond to their perceived obligations as family[13].

To the best of my knowledge, there have been no tests of the hypothesis that kin density declines as small-holder farmers transition to a fully market-dependent economy[61]. Here I address this gap by analysing the ego-networks of 1995 women in 22 communities in an area of rural Poland characterised by centuries of peasant subsistence farming but which has been abandoning these practices since Poland's accession to the European Union in 2004[68]. This mid-transitional context allows: (1) a sampling of populations on a spectrum of market integration; (2) a measure of market integration that can be explicitly compared across

communities and levels of analysis; (3) control for cultural, religious and linguistic differences that might independently structure personal networks; and finally (4) a new, explicit measure of kin density. In previous work in this region I and my colleagues have examined how the transition to market dependence may be altering women's reproductive strategies[32,68] via personal networks that channel contraceptive use[46] and low fertility values[69] and through contextual effects at the community level[46,69]. Here I show that a key mechanism—declining kin density of personal networks—is also evident during this transition.

## Results

**Measuring market integration.** At the time of the survey (2009–2010), more than 65% of participants lived in households subsisting on farming, with additional incomes combining formal and informal wage labour, seasonal and migrant employment[68,70]. Working patterns[28] are shifting steadily towards the labour market as the population turns away from traditional inheritance and post-marital migration strategies[68] and as fertility declines. To capture market integration I created a weighted composite index of household members' occupation, occupational prestige and employment history (up to 15 people, see[32,69]). The diversity of income sources allows the incorporation of both 'traditional' and 'modern' dimensions of wealth creation[32]. Market integration is standardised and then group-mean centred within communities: an individual's score is her household's deviation from the community mean.

**Demographic and network features of the study communities.** The study communities are saturated with kin (Fig. 1). This is due to the combined effects of: (a) high completed fertility (mean 3.8, s.d. 2.15. Note the Polish total fertility rate when the data were collected was 1.2) and low levels of lifetime childlessness (~5% of all 907 post-reproductive women remained childless); (b) high

levels of endogamous marriage (28% of marriages were contracted within the community, ranging from ~5 to 54% across villages); (c) relatively late age at out-migration for women who moved (mean 24.5 years, s.d. 9.2); (d) short distances between natal and post-marital residences, for both men and women (modal distance is 2–3 km); and (e) strong between-community marriage connections (75% of all marriages, $n = 1205$, were contracted within the 22 study communities).

During semi-structured interviews, all women (aged 18–91, mean 44 years, s.d. 17.8) were prompted with the following name-generator: 'Please name up to five women you could call your friends. These are women you are close to and who you can talk to about important personal matters (for example about children, family, health or other things). These women can be your relatives or come from outside your family. They can be from this village or anywhere else. You don't have to name five.' (see Methods, Supplementary Note 1 and Supplementary Table 1 for more details). A five alter limit was imposed to capture core social support partners. Fewer than 20% of participants ($n = 386$) used all five nominations. Ego then reported on the alter characteristics and their connections (e.g. consanguineal or affinal kin. See Supplementary Table 1). The networks are balanced between kin (48% of all alters) and non-kin (52%), and geographically bounded: 53% of alters live in the same community as ego, 49% come from the same community. These are predominantly long-term, active relationships. Mean length of friendship is 23 years (s.d. 13.6). 43% of ego-alter ties see each other daily, and 66% of ego-alter ties talk on the phone daily.

**A new measure of kin density.** Previous research has examined changes in the proportion of kin in ego-networks, but not the density of kinship connections (i.e. the proportion of possible kinship connections between alters). These are different measures that should be distinguished. Two networks with the same proportion of kin (0.6, or 3 kin members in a five-person ego-

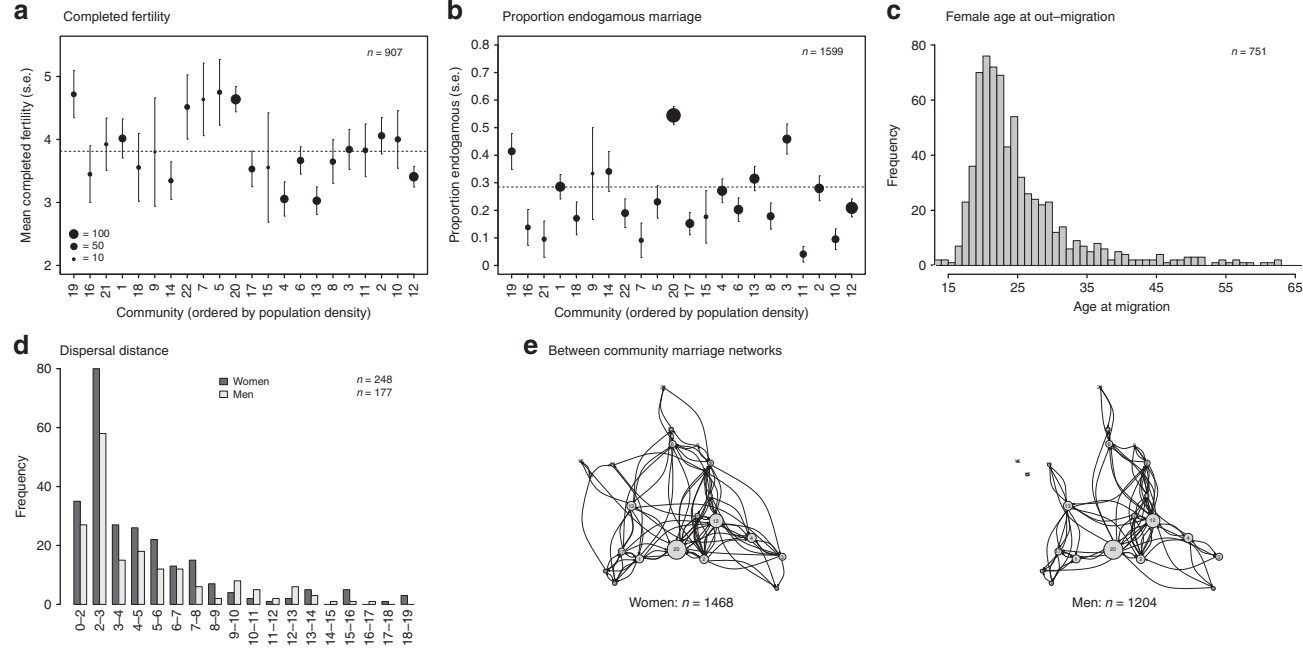

**Fig. 1 Demography of the study communities.** Between-community variation in **a** completed fertility (women aged 45+), and **b** proportion of endogamous marriages (means are indicated by dashed lines); **c** age at out-migration for women; **d** frequency distribution of distances between natal and post-marital residence for men and women, and **e** marriage connections between communities for women (left) and men (right). Communities in **a**, **b** are ordered from left to right by increasing population density. Communities in (e) are shown in relative geographic space: edge width indicates the frequency of people moving to a particular community after marriage. Data are provided in a Source Data file.

## Network density

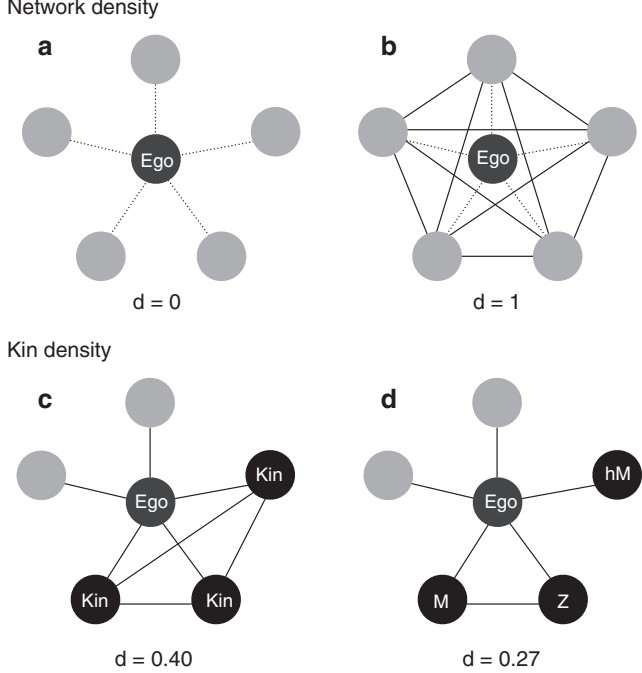

**Fig. 2 Differences between network density, kin density and the proportion of kin. a** shows a sparse ego-network ($d = 0$, i.e. no ties present) and **b** a fully dense one ($d = 1$, i.e. all possible ties present). Ego's connections to the alters are not counted (dashed lines). **c** shows a moderately kin-connected network ($d = 0.4$, i.e. 40% of all possible ties are kinship ties), where the proportion of kin is 0.6. Ego's kinship connections to the alters are counted. **d** shows kinship density for the same network when affinal kin (hM = husbands mother) are distinguished from consanguineal kin, reducing the number of kin ties (M = mother; Z = sister; $d = 0.27$, i.e. only 27% of ties are kinship ties). Counting the proportion of kin in the network is not equivalent to counting the density of kin connections.

network) can differ in kin density depending on how many affinal versus consanguineal kin ego nominates. If ego nominates her mother (**M**), sister (**Z**) and husband's mother (**hM**), kin density will be 0.27 (**M** and **Z** are kin, but neither **M** nor **Z** have a kinship connection with **hM**). If ego nominates three consanguineal kin, all of whom are related, this value will be 0.40 (see Fig. 2). If ego replaces consanguineal ties with affinal ties after marriage (see Fig. 2), the kin density of her network will actually decline even though the proportion of kin may not. This distinction is important because there is an implicit assumption that more kin translates directly into larger 'kin effects' on reproduction. As a proxy for social interaction and information transmission, kin density is more explicit about which kin are (or are not) connected, and therefore better captures the likelihood of compound or 'coordinated' interests in, and effects on, reproductive outcomes.

**Multilevel analysis of network characteristics.** In three separate multilevel models, I examine associations between household and community level market integration and network: (M1) size: the number of alters ego nominates; (M2) density: the proportion of all possible ties between alters in the network that are present, and (M3) kin density: the proportion of all possible ties that are kinship connections (Fig. 2). Ego-alter connections are excluded from the density measure as every alter is necessarily connected to ego. A network with two alters has a density of 0 (unconnected) if alters do not know each other and a density of 1

(closed) if they are connected independently of ego. Ego's kinship connections to alters are counted for kin density because ego is not necessarily related to every alter.

**Descriptive statistics for size, density and kin density.** Figure 3a shows that mean ego-network size is 2.92 (s.d. 1.46). It is 3.10 (s.d. 1.30) when excluding the 118 respondents who did not nominate any friends. These values are consistent with networks among subsistence horticulturalists[45] and in the USA[60]. Mean density is very high at 0.80 (s.d. 0.32, i = 1637), i.e. 80% of possible ties between alters are present on average. Mean kin density is much lower overall, at 0.38 (s.d. 0.35, $n = 1854$), i.e. 38% of possible ties (including those between ego and alters) are kinship ties. Density and kin density are moderately positively correlated (Pearson's $r = 0.14$, 95% CI [0.10, 0.19]). There is a small negative correlation between density and network size (Pearson's $r = -0.09$, 95% CI [−0.14, −0.04]), but no correlation between kin density and network size (Pearson's $r = -0.02$, 95% CI [−0.07, 0.02]). In other words, bigger ego-networks tend to be a little less dense, but they do not tend to be less kin-dense.

Mean ego-network size in a community does not tend to co-vary with community market integration (Fig. 3a). However density, and more strongly, kin density, tends to be lower in communities with higher market integration (Fig. 3b).

**Kin ties loosen but networks do not shrink.** As shown in Fig. 3c, the multilevel models reinforce this broad pattern: that women living in more market-integrated communities are expected to have lower density and kin density, but not smaller network size (Supplementary Table 2). Independent of other variables, a 1 s.d. increase in community market integration is associated with 15% lower odds of there being connections between alters in the network ($\beta = -0.16$, s.e. = 0.07, OR = 0.85, 95% CI [0.74, 0.98]) and 13% lower odds of there being kinship connections between them ($\beta = -0.14$, s.e. = 0.05, OR = 0.87, 95% CI [0.79, 0.95], Fig. 3c and Supplementary Table 2).

For kin density, but not density, the association with market integration is also apparent at the household level: a one-unit (~1 s.d.) increase in household market integration is associated with 11% lower odds of kinship connections being present between alters in the network ($\beta = -0.11$, se = 0.02, OR = 0.89, 95% CI [0.85, 0.94], Fig. 3c and Supplementary Table 2). So both household and community level market integration are associated with reduced kin density, with the effect size being larger at the community level. This means that irrespective of how market-integrated a woman's own household is, when others in her community are highly market-integrated, her network is substantially less kin-dense than would otherwise be expected.

**Individual characteristics matter more for density.** Three individual level characteristics are importantly associated with density: whether or not the respondent was married, a farmer, or a migrant. For kin density, only marital status is associated. None of these variables are strongly associated with network size (see Fig. 3c and Supplementary Table 2). Married women have 63% higher odds of their alters being connected than unconnected, compared to unmarried women ($\beta = 0.49$, s.e. = 0.10, OR = 1.63, 95% CI [1.34 1.99]) and they have 2.3 times the odds of alters having kinship connections ($\beta = 0.84$, s.e. = 0.08, OR = 2.32, 95% CI [2.00, 2.69]). Women who are farmers have 20% higher odds that alters in their networks are connected compared to non-farmers ($\beta = 0.18$, s.e. = 0.07, OR = 1.20, 95% CI [1.04, 1.38], see Fig. 3c and Supplementary Table 2), but the odds of kinship connections between alters are no different ($\beta = 0.05$, s.e. = 0.05, OR = 1.05, 95% CI [0.95, 1.16], see Fig. 3c and Supplementary

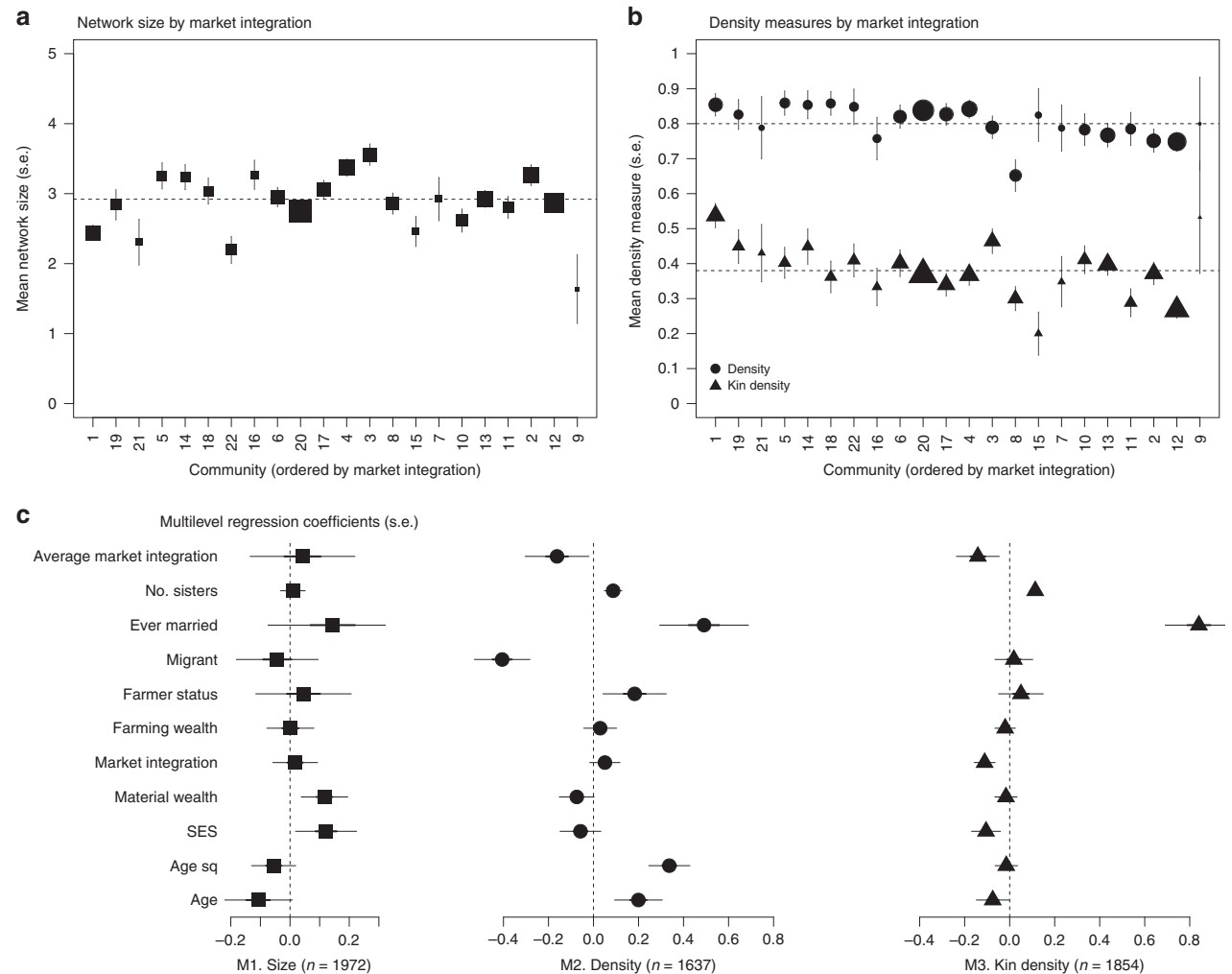

**Fig. 3 Community level variation in network characteristics.** Panels show **a** network size and **b** density/kin density, ordered by average market integration (see Source Data file). **c** shows forest plots of the model-adjusted coefficients (ß estimates) and standard errors from the three multilevel models (see Supplementary Table 2).

Table 2). And migrant women have 33% lower odds that their alters are connected compared to women staying in their home communities ($\beta = -0.41$, s.e. $= 0.06$, OR $= 0.67$, 95% CI [0.59, 0.75]), but again the odds of kinship connections are no different ($\beta = 0.02$, s.e. $= 0.04$, OR $= 1.02$, 95% CI [0.94, 1.11], see Fig. 3c and Supplementary Table 2). So married women, farmers and 'stayers' are expected to have more dense ego networks, but only married women are expected to have more kin-dense networks.

Socio-economic status was associated with reduced kin density, with a one-unit (~1 s.d.) increase associated with 10% lower odds of there being kinship connections between alters ($\beta = -0.11$, s.e. $= 0.03$, OR $= 0.90$, 95% CI [0.84, 0.96], Fig. 3c and Supplementary Table 2). This is not true for density. High socio-economic status women also tended to nominate slightly more friends, as did materially wealthy women ($\beta = 0.12$, s.e. $= 0.05$ and $\beta = 0.12$, s.e. $= 0.04$ respectively). Finally, for every additional sister a woman has, the odds of her alters being connected are 9% higher ($\beta = 0.09$, se $= 0.02$, OR $= 1.09$, 95% CI [1.05, 1.13]), and the odds of kinship connections are about 12% higher ($\beta = 0.11$, se $= 0.01$, OR $= 1.12$, 95% CI [1.09, 1.15]).

**Subset analyses support the results for kin density.** Subset analyses indicate that various compositional features of the communities do not drive the associations between increasing market integration and declining kin density. Among the subsets of married women ($n = 1598$), farmers ($n = 1239$), women who stayed in their home communities ($n = 1221$) and women with either one or no sisters ($n = 976$), increasing community and household market integration are strongly associated with reductions in kin density (see Supplementary Tables 3–6). In every subset, the magnitude of the association between market integration and kin density is comparable to or larger at the community than at the household level. Among married women there is a community level association between increasing market integration and declining density, but there are otherwise no strong or consistent associations between market integration and density in any of the subsets.

**Other community-level predictors do not alter the results.** It is not possible to distinguish community-level market integration from population density in these data, since they are strongly correlated (Pearson's $r = 0.60$, 95% CI [0.57, 0.62]). Replacing market integration with population density in the models yields qualitatively similar results (see Supplementary Table 7), with the disadvantage that we lose the explicit multilevel comparison: we cannot meaningfully compare a unit increase in population density with a unit increase in household market integration. However other community level variables can possibly be ruled

out as primary drivers of these results. Although community-level fertility and migration rates might make more/fewer kin available as network partners, including average completed fertility or proportion of migrants in the models does not alter the results (see Supplementary Tables 8 and 9). Ultimately, a larger sample of communities may be needed, as well as formal models, to tease apart these factors. It is important to note that these data are cross-sectional and the analyses correlational, so causality cannot be established. Nonetheless, the counterfactual—that declining kin density itself causes increased market integration—seems a less plausible account of the relationships in these communities.

## Discussion

The data and analyses presented here fill an important empirical gap in our understanding of how social networks change in the course of market integration. They provide: a new measure, kin density, distinguished from density and the proportion of kin; comparison of network characteristics across multiple communities of the same ethno-linguistic and religious group, and explicit comparisons of market integration using the same units at different levels of analysis. Across 22 farming communities in rural Poland—representing a mid-transition context—personal networks are somewhat less dense, but much more clearly, less kin-dense in more market-integrated communities and households. This apparent 'loosening' of kin connections is occurring despite the fact that ego-network size does not decline with market integration, and is independent of a range of individual-level characteristics. This corroborates a widespread assumption about how social interactions change as populations 'modernise'[41] but which has not, to my knowledge, been explicitly demonstrated before. The results suggest that kin density may decline more strongly than density, or perhaps prior to it, in the later stages of market integration.

Consistent with other evidence, married women have higher kin density, so they are either building new kinship connections or are simply more reliant on kin following marriage. There is a strong emphasis on affinal kin (particularly the husband's mother) in theorising about how kin influence women's fertility[71], child mortality[40] and contraceptive uptake[47,52]. Carefully distinguishing kin density from the proportion of kin allows relationships between different family members to be accurately accounted for, and avoids overestimating kin effects by recognising that affines are often unrelated to the other kin nominated by ego. This matters because of an implicit assumption that kin have compound effects on reproduction: the more kin in the network, the stronger the kin influence. Future work should expand kin density to incorporate weighted kinship ties, which could further delineate a variety of relations among kin.

When living in market integrated communities, married women, farmers, women staying at home and women with few or no sisters had 'looser' kinship connections than their individual characteristics would suggest. This speaks to the broader influence of higher-level socio-cultural features bearing on their social interactions. Certain compositional features of the study communities—the proportion of migrants, the local fertility levels—do not seem to be the main drivers. We can appeal to mechanisms at the macro-level, but it remains unclear how to connect these to the micro-level features described here. Broad changes in social transmission, brought about by market integration via mass media, new workplaces, schooling, and the increased mobility associated with accessing these[3], may accelerate both the speed at which friendships become less kin-oriented and the level of exposure to new ideas outside the kin-group. Macro-level fertility decline in principle reduces the number of available kin in the broader population[72,73], but local fertility levels were unimportant here. Rates of in- or out-migration due to marriage, employment or educational opportunities may subtly alter availabilities of kin, though again, having more migrants in the community does not seem to play an important role at the level of aggregation studied here.

The above ideas all assume that kin are 'lost' in similar ways in different contexts. But declining kin prominence depends on the context-specific support they can provide[7,74], and their ability to adapt support strategies to changing circumstances. In urban China[61] kin were nominated less over time because ego's needs were better fulfilled by social contacts outside both the family and the workplace. Under state socialism, co-workers with job security could secure basic economic and social resources that kin could not offer, crowding them out. This implies a facultative restructuring of personal friendships to meet the demands of economic change. An analogous restructuring may be emerging in the post-socialist context of the Polish study communities. Following centuries of peasant farming[75], the area was not collectivised under socialism (1945–1989), and traditional lifeways remained intact during the dramatic and destabilising formal transition to market economy (1989–1991). Since accession to the EU (2004) the government has enacted reforms to 'modernise' peasant farming, incentivising smallholders to give up[76]. Until very recently, the combination of low mobility, high fertility, short migration distances, male-biased inheritance practices[68], cooperative farm work and childcare generated dense marriage networks and a shared connection to the land. Now, decreasing fertility[69] and shifting inheritance strategies that distribute parental investments between children's education and farming—and which directly influence migration patterns[68]—are transforming the social environment towards a less land- and kin-oriented form.

Why should market integration be associated with this effect? First, market economies allow for clearer separations between the domains of family and work than in subsistence economies, and that may let kin relations fray. Second, emotional support and companionship may be more explicitly valued under market economies than material and physical help, further emphasising the demand for self-similar (homophilous) network partners over kin. Third, material support is more likely on the scale of larger, less-frequent transfers in market-dependent economies than the small and frequent transfers typical in less market-dependent ones, and this may also decrease the frequency of kin interactions. Fourth, fungible assets (such as money) that can be immediately exchanged, saved for future expenditures, and used to pay off debts instead of providing in-kind services or other goods may also allow a greater distance between kin while loosening obligations to collective activities. Of course, the process may not be smooth. Among Pimbwe communities in rural Tanzania, inequality in market integration alongside perceived declines in self-sufficiency[74] generated fragmented and even antagonistic kin relations. On the other hand, among Tsimane horticulturalists in Bolivia, wealthier, more educated individuals living in market-integrated communities shared food more and invested more of their labour in collective activities, maintaining their traditional food-sharing networks[33]. In the Polish context, advanced market integration and education have changed what counts as wealth and status and how they are translated into reproductive strategies[32]. Some communities are more unequal as this process unfolds[32], but with farming still a viable livelihood, cooperative activities remain important and this work is still preferentially carried out with kin. There remain impediments to income diversification, including lack of opportunities or accumulated human and material capital[77]. There is evident tension between a desire to maintain a traditional lifestyle and the perceived benefits of the market economy. For example, 78% ($n = 1455$) of

interviewees expressed a wish to live in a village rather than a town but only 48% said they wanted their family to continue farming in the future[70].

Despite decades of research on fertility decline, the social dynamics of reproduction are still poorly understood. We do not yet know if fertility outcomes are directly or indirectly influenced by reduced kin contact. Declining kin interactions might be an epiphenomenon of broader structural changes (wage-labour makes it convenient or expedient to interact with non-kin if they are important conduits to resources), or demographic ones (fewer kin available due to declining fertility). Either way, fragmenting kin networks potentially allow the percolation of values unconnected to reproduction through the population[41] and this important cultural mechanism has received little attention. The adoption of norms and values associated with low fertility, driven by a range of social transmission biases (such as conformity), may depend on a prior loosening of these kinship ties.

Global fertility decline already has and will continue to have enormous impacts on demography and cultural change in the future. Connecting social networks to local demography is crucial for a better understanding of the mechanisms underlying this transition. The results here should help guide and provide parameters for formal models that are now needed to establish causal connections. Future work should examine whether small differences in the composition and density of ego networks scale up to: (a) change the balance of information relevant to reproduction; (b) alter the rates and channels by which reproductively relevant information is transmitted and (c) feed-back to transform reproductive preferences and rates[78].

## Methods

**Data collection**. The study was approved by the Ethics Committee of the Department of Anthropology at UCL. The data reported in this study were collected between 2009 and 2010 in rural southern Poland (sampling information can be found in the Supplementary Note 1). All participants gave informed consent. The section of the questionnaire used to elicit the social network data used in this study is provided in English and Polish in the Supplementary Note 1. All social network information was provided by ego, so the reported ties are undirected. While it is assumed that husbands/partners and male kin are important in the personal networks of respondents, the aim of this study was to examine how other women influenced the respondent's behaviour, so data were not collected on male network partners. These networks should be seen as support networks most relevant to the domain of reproduction. Supplementary Table 1 gives an overview of their main characteristics.

**Covariates in the multilevel models**. Covariates in the models are: farmer status (respondent is resident in a farming household at the time of the survey or not; n 'farmers' = 1239, n 'non-farmers' = 733), age of the respondent (mean = 44, s.d. 17.83, range 18–91), age squared to account for the non-linear relationship between network characteristics and age across the lifespan, marital status (n 'ever married' = 1598, n 'never married' = 374), migrant status (n 'migrants' = 751, n 'non-migrants' = 1221), number of sisters (mean = 1.85, s.d. = 1.55, range 0–8), socio-economic status, material weath and farming wealth. The latter three covariates are weighted linear combinations of multiple variables, created using Principal Component Analysis (PCA) and then standardised to have a mean of 0 and s.d. of ~1[32]. Socio-economic status was created using the following variables: education level of the respondent and both of her parents, parental employment history (i.e. whether the parents were ever engaged in wage labour or not), and whether the main income in the respondent's childhood was derived from farming or wage labour. Parental employment history is a good indicator of socio-economic status in this population, since relatively few people would have been engaged in waged employment in previous generations. Since it is known that socio-economic status is related to the characteristics of parents as well as those of the focal individual, this measure constitutes a more long-term definition than education of the respondent would on its own. Material wealth was created using the following variables: mean household income (measured across all adults in the household in brackets ranging from <600 Polish Zlotys [PLN] per month, to >2500 PLN per month), ownership of a computer, an internet connection, a satellite TV, a car, as well as the number of habitable rooms in the house. This captures both income and assets, and covers the whole household rather than simply the parents or the husband of the respondent. Farming wealth was created using the following variables: the number of livestock and amount of land owned, ownership of farm equipment, and indicators of whether the family earns any income from

farming besides subsistence. Market integration captures the extent to which the household is dependent on wage-labour versus farming income. For every householder, occupation (ranging from student to farmer to full-time employed), occupational status (if employed) and a binary indicator of employment history (ever employed) were weighted to provide a cumulative score. I took the household mean as the measure of overall market integration. The respondent is not included in this measure to avoid confounding co-linearity with the other measures, specifically individual education. Further details of the construction of these variables are available in Colleran et al.[32] and Colleran[70].

Market integration, socio-economic status, material wealth and farming wealth are all standardised to have mean = 0 and s.d. = 1, and then group-mean centered: a woman's score in the analyses is her deviation from the community mean. Group-mean centering allows market integration to be used as a community-level predictor without introducing confounding collinearity with the household level variable (see[32]). Un-centered versions of this variable produce qualitatively identical results (see Supplementary Table 10). Age is standardised to have mean = 0 and s.d. of 1 (equivalent to 17.84 years), centered on the grand mean: a woman's score is her age deviation from all other women in the sample (n = 1972). The three binary variables included in the analysis—farmer status, marital status and migrant status—are untransformed. Number of sisters (ranging from 0 to 8) is also an untransformed continuous variable (see Supplementary Note 1 for further details).

**Multilevel statistical analysis**. All analyses were carried out in R version 3.6 using the 'lme4'[79] 'arm'[80] and 'blme'[81] packages. Multilevel regression models use either linear (network size) or binomial (density, kin density) error distributions. Density and kin density are entered as a two-column matrix of 'successes' (i.e. number of possible connections present) and 'failures' (number of possible connections absent). Regression estimates, transformed into odds ratios, give the odds of connections between alters relative to the absence of connections. All models were estimated using either restricted maximum likelihood (REML) for linear models or maximum penalised likelihood (MPL)[82] with the Laplace approximation for the binomial models. MPL uses weakly informative priors to obtain Bayesian modal estimates on the parameters in the model, to avoid boundary problems associated with small variance components and a small sample of groups, including incorrect or underestimated uncertainty in the model parameters and covariance matrices. This ensures that the variance component estimates remain off the boundary of the feasible parameter space (zero), but with weak enough priors so that inferences remain consistent with the data. This method extends standard multilevel modelling techniques without requiring simulation as in fully Bayesian analysis (which obtains posterior mean estimates) and outperforms standard maximum likelihood methods. Network plots were produced using the iGraph package.[83]

Twenty-three women not currently living in any of the communities were removed from the analysis of network size, leaving n = 1972. For density, only respondents who named at least two friends were included, leaving n = 1637. For kin density, individuals not nominating any friends were excluded, leaving n = 1854.

**Reporting summary**. Further information on research design is available in the Nature Research Reporting Summary linked to this article.

## Data availability

In order to protect the privacy of participants, the data used in this study is not publicly deposited, however the data that support the findings are available from the author upon request. Source data for Figs. 1, 3a, b are provided as a Source Data file. Source data for Fig. 3c can be found in Supplementary Table 2.

## Code availability

The R code that supports the findings of this study is available from the author on request.

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

## Acknowledgements

I am indebted to the women of Limanowa district who took part in this study. I thank Grazyna Jasienska, Andrzej Galbarczyk, Ilona Nenko and Ruth Mace for support during the study and my field assistants from all stages in the project, the Wojtas and Markiewicz families, and local parish rectors. This work was funded by the ESRC, Wenner-Gren Foundation (dissertation fieldwork grant 8182), UCL Graduate School and Gay Clifford Fund. I thank Adam Powell and Anne Kandler for comments on the paper.

## Author contributions

H.C. designed the study, collected the data, performed analyses, and wrote the paper.

## Competing interests

The author declares no competing interests.
