## [Peer Review File · Nature Communications]

Reviewers' Comments:

Reviewer #1:

Remarks to the Author:

This is an interesting paper that looks at the size and kinship structure of women's personal social network in the transition from a peasant market economy to a full market economy in contemporary Poland. We have very few studies that have explored this key topic, especially in European communities. So this paper is very welcome. The analyses are fine, and the dataset is very substantial. My only quibble is that I think the paper is mis-framed to imply that it is examining the transition from a subsistence economy to a full market economy. It isn't, and that just needs resetting.

I have only a few comments.

l. 11: I am not aware of any suggestion that they shrink. Citation?

l. 21: wording is awkward here: market integration cannot 'have' anything, it can only result in something [or women that are market integrated can have networks....]

l. 31-5, 47-48 and 63-64: this all makes it seem as though this study will redress this balance by presenting data on a small scale traditional society. It was not until the last paragraph of the Discussion that I actually realised that this study was carried out in Poland (l. 157 notwithstanding)! This is not a transition from a subsistence economy to a market economy: this has been imbedded in a market economy (one based on the use of money at least some of the time) for 2000 years. It is a transition from a peasant market economy (peasants sell surpluses when they have them, and always have) to a full market economy, and this needs to be made much clearer right up-front. It would help if the location was stated in the Abstract.

l. 407-8: shouldn't this be evident from the data? In many ways, this is crucial because of the impact that reduced family size has on network structure (see David-Barrett & Dunbar, 2017, *J. Theoret. Biol.*). And you surely must know.

Reviewer #2:

Remarks to the Author:

This paper reports an analysis of a study of the close networks of women in 22 Polish farming communities. They are in a region that had remained in relative economic isolation far longer than most parts of Europe and the mean fertility of the women in these communities was considerably higher than the European average. An association between high fertility and low market integration has been observed in many populations and there has been much discussion of possible mechanisms by which increasing market integration might cause a reduction in fertility. At the time of the study (2009-10), these communities were becoming more market oriented and the degree of market integration varied between communities. This population provided an opportunity to look in detail at the effects of market integration. The study shows that market integration is associated with a reduction in the kinship density of women's networks.

The study was well-conceived and carried out. We do not claim expertise in social network analysis, but the analysis seems thorough and appropriate. The paper explains the findings clearly and includes a useful review of the literature which suggests that looking at variation in the composition and structure of the social networks in the communities is likely to yield insight.

The reproductive norms of modern societies are very different from those which existed for most of human evolutionary history. They continue to evolve rapidly and in complex ways with many coevolving factors. The results reported in this paper suggest that all this complex cultural evolution may depend on a prior loosening of kinship ties which, for most of human evolutionary history, had kept humans bound into families which competed for survival and reproductive success.

Minor suggestions:

1) The introduction of this paper would seem slightly more complete if the widely cited 1971 paper by Wilbur Zelinsky was briefly mentioned: Zelinsky, W. . 1971. 'The hypothesis of the mobility transition', *The Geographical Review*, 61: 219-49.

2) The paper would might be more likely to attract attention outside academia if the introduction pointed out that the cause of the fertility decline really is a mystery. The explanations that are widely believed and still taught are not supported by evidence.

A brief reminder of the huge impact the fertility decline is having and will continue to have on demography and the evolution of cultures might also be useful.

3) Monique Borgerhoff Mulder is cited two different ways in this paper. We believe she prefers Borgerhoff Mulder, M. to Mulder, M.B.

Lesley Newson and Peter J. Richerson

Response to Reviewers' comments:

Reviewer #1 (Remarks to the Author):

This is an interesting paper that looks at the size and kinship structure of women's personal social network in the transition from a peasant market economy to a full market economy in contemporary Poland. We have very few studies that have explored this key topic, especially in European communities. So this paper is very welcome. The analyses are fine, and the dataset is very substantial. My only quibble is that I think the paper is mis-framed to imply that it is examining the transition from a subsistence economy to a full market economy. It isn't, and that just needs resetting.

Thank you for the general point, which I have taken onboard. I have gone through the paper thoroughly for problematic framing and hope it no longer gives this impression.

I have only a few comments.

l. 11: I am not aware of any suggestion that they shrink. Citation?

There is good evidence that personal networks have been shrinking over recent decades in the USA, and this was the work I was referring to in the original submission. However, the same limitation applies here as does with the other sociological research cited in the paper, in that it was not carried out in the kind of population relevant for my question. For that reason, I have removed reference to "shrinking" networks in the abstract, but the evidence is still mentioned on lines 90-93.

l. 21: wording is awkward here: market integration cannot 'have' anything, it can only result in something [or women that are market integrated can have networks....]

Quite right. I have rephrased this wording throughout the paper.

l. 31-5, 47-48 and 63-64: this all makes it seem as though this study will redress this balance by presenting data on a small scale traditional society. It was not until the last paragraph of the Discussion that I actually realised that this study was carried out in Poland (l. 157 notwithstanding)! This is not a transition from a subsistence economy to a market economy: this has been imbedded in a market economy (one based on the use of money at least some of the time) for 2000 years. It is a transition from a peasant market economy (peasants sell surpluses when they have them, and always have) to a full market economy, and this needs to be made much clearer right up-front. It would help if the location was stated in the Abstract.

Thank you for pointing this out. I have now made it clear both in the abstract and throughout the paper where the study population is. I have also removed references to "small-scale" wherever possible to avoid this misunderstanding, and have contextualized the Polish context more to reflect your concerns.

l. 407-8: shouldn't this be evident from the data? In many ways, this is crucial because of the impact that reduced family size has on network structure (see David-Barrett & Dunbar, 2017, J. Theoret. Biol.). And you surely must know.

Thank you for the reference. If I understand the comment correctly, the thing that should be evident is that women who come from smaller families in the dataset should have fewer kin available and therefore lower kin density in their network. That's true to some extent: women with more sisters do have more kin-dense networks (lines 248-250). It is tricky to go directly from this to saying that fertility decline is the thing that drives the association between market integration and reduced kin density in ego-networks. There are two reasons I make this claim: (1) among women with one or no sisters, the relationship between market integration and declining kin-density still holds (lines 257-259) and (2) including mean completed fertility as a proxy for varied fertility rates does not alter the results (lines 271-274). I write in the discussion (lines 309-320) that of course, rates of fertility decline and migration at higher levels of aggregation may be affecting these relationships, but that it is not clear at present how to connect these macro patterns with the micro data in my study.

Reviewer #2 (Remarks to the Author):

This paper reports an analysis of a study of the close networks of women in 22 Polish farming communities. They are in a region that had remained in relative economic isolation far longer than most parts of Europe and the mean fertility of the women in these communities was considerably higher than the European average. An association between high fertility and low market integration has been observed in many populations and there has been much discussion of possible mechanisms by which increasing market integration might cause a reduction in fertility. At the time of the study (2009-10), these communities were becoming more market oriented and the degree of market integration varied between communities. This population provided an opportunity to look in detail at the effects of market integration. The study shows that market integration is associated with a reduction in the kinship density of women's networks.

The study was well-conceived and carried out. We do not claim expertise in social network analysis, but the analysis seems thorough and appropriate. The paper explains the findings clearly and includes a useful review of the literature which suggests that looking at variation in the composition and structure of the social networks in the communities is likely to yield insight.

The reproductive norms of modern societies are very different from those which existed for most of human evolutionary history. They continue to evolve rapidly and in complex ways with many coevolving factors. The results reported in this paper suggest that all this complex cultural evolution may depend on a prior loosening of kinship ties which, for most of human evolutionary history, had kept humans bound into families which competed for survival and reproductive success.

Minor suggestions:

1) The introduction of this paper would seem slightly more complete if the widely cited 1971 paper by Wilbur Zelinsky was briefly mentioned: Zelinsky, W. . 1971. 'The hypothesis of the mobility transition', *The Geographical Review*, 61: 219-49.

Many thanks for reminding me to include this paper, which I had forgotten. I agree it is essential and it is now cited throughout the manuscript.

2) The paper would might be more likely to attract attention outside academia if the

introduction pointed out that the cause of the fertility decline really is a mystery. The explanations that are widely believed and still taught are not supported by evidence.

I agree completely. I have refocused the writing to try to bring these issues more to the fore.

A brief reminder of the huge impact the fertility decline is having and will continue to have on demography and the evolution of cultures might also be useful.

Again I completely agree, and thank you for the comment. I have added some new text to this effect in both the introduction and conclusion sections of the paper.

3) Monique Borgerhoff Mulder is cited two different ways in this paper. We believe she prefers Borgerhoff Mulder, M. to Mulder, M.B.

Duly noted, many thanks.

Reviewers' Comments:

Reviewer #2:

Remarks to the Author:

The author successfully dealt with all of our concerns. This manuscript is ready for publication.